# 3-D Modeling of Gas–Solid Two-Phase Flow in a π-Shaped Centripetal Radial Flow Adsorber

**Haoyu Wang** [1,2,†], **Xiong Yang** [3,†] 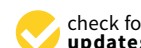, **Ziyi Li** [3], **Yingshu Liu** [3], **Chuanzhao Zhang** [1,*], **Xiaojun Ma** [1] **and Chunwang Li** [1]

[1] College of Biochemical Engineering, Beijing Union University, Beijing 100023, China; wanghaoyu81@126.com (H.W.); maxiaojun@buu.edu.cn (X.M.); jdtchunwang@buu.edu.cn (C.L.)

[2] Department of Agricultural and Biological Engineering, University of Illinois at Urbana-Champaign, Urbana, IL 61801, USA

[3] School of Energy and Environmental Engineering, University of Science and Technology Beijing, Beijing 100083, China; yangx@ustb.edu.cn (X.Y.); ziyili@ustb.edu.cn (Z.L.); ysliu@ustb.edu.cn (Y.L.)

[*] Correspondence: Chuanzhao.zhang@163.com; Tel.: +86-10-52072257

[†] These authors contributed equally to this work.

**Featured Application: A 3-D modeling of gas–solid two-phase flow can better reflect the actual PSA oxygen production process. The laws of the pressure, velocity, temperature and oxygen distribution in the adsorber could provide an important technical reference for CP-π RFA in the PSA for oxygen production.**

**Abstract:** Radial flow adsorber (RFA) is widely used in large-scale pressure swing adsorption (PSA) oxygen production system because of high air separation. In this study, a 3-D modeling of gas–solid two-phase flow was established for the π-shaped centripetal RFA (CP-π RFA). The pressure difference, temperature changes, velocity profiles and oxygen distributions were comparatively studied using this model. Part of the results have been compared with the experiments results, which shows this model can give an accurately prediction. The results show that the pressure and velocity in the adsorber change greatly near the outer hole and central hole, but the overall pressure and velocity changes in the bed are stable. The oxygen product purity in the adsorbent filling area performed better on oxygen enrichment after eight cycles. The oxygen product flow rate will affect the oxygen production performance. The laws of the pressure, velocity, temperature and oxygen distributions can provide an important technical reference for CP-π RFA in the PSA for oxygen production.

**Keywords:** radial flow; π-shaped centripetal; adsorption; CFD modeling; two-phase flow

## 1. Introduction

The most typical pressure swing adsorption (PSA) process is a two-bed four-step Skarstrom cycle, the schematic is shown in Figure 1 [1]. The cyclic process consists of pressurization, adsorption, countercurrent blow down and purge. It is the basis for designing more complex PSA processes [2]. Currently there are two-bed PSA, three-bed PSA, four-bed PSA and vacuum pressure swing adsorption processes. The PSA oxygen production system has been widely used in industrial production due to its low energy consumption, simple equipment, and convenient operation. However, with the continuous increase of air processing capacity and the decrease of the utilization efficiency of adsorbent, oxygen production system is seriously restricted by the development of large-scale and low energy consumption. Conventional large-scale axial flow adsorbers have large void volume percentages, and uneven flow distribution, which will seriously affect the oxygen production [3]. In contrast,

radial flow adsorber (RFA) has the characteristics of large cross-sectional area, large flow capacity, gas distribution uniformity, small flow resistance and large feed flowrate [4,5]. It has been widely used in chemical industry, petrochemical and gas production industry [6,7]. However, due to the extremely complex variable mass flow caused by the internal structure of RFA, one of the most important issues to be solved for heat and mass transfer in two-phase flow in RFA is model verification [8]. LaCava et al. [9] pointed out RFA designs were becoming more popular because of high velocities and short mass-transfer zones.

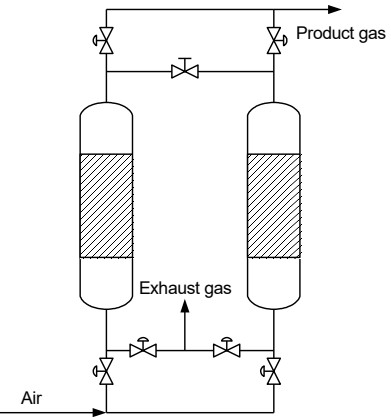

**Figure 1.** Schematic of the Skarstrom cycle.

Kareeri et al. [10] studied the flow distribution of radial-flow beds using computational fluid dynamics and showed that the centrifugal-π type had a significant uniform flow. Celik [11] proposed to improve the uniform distribution of airflow by using the non-uniform opening distribution of the inner and outer cylinders. Zhang et al. [12] developed a differential equation for the Z-flow type RFA by pressure drop analysis considering the change in the flow area of central pipe. The ideal shape of the central pipe along the axis was obtained by solving the differential equation, which could be realized by inserting a cone with a variable cross-sectional radius along the axis in the middle of the central pipe. Farsi [13] investigated one and two-stage spherical reactors through a mathematical model. To verify the accuracy of the considered model, the simulation results of a conventional reactor were compared with the plant data and found that DME (Dimethyl ether) production was improved by about 2.8% and 3.7% in the one and two-stage membrane configurations against conventional tubular reactor. Li et al. [14] measured the pressure profiles in the flow channels and layers of a three-layer RFA and investigated the properties of the fluids in flow channels of a three-layer RFA. Zheng et al. [15] established a two-dimensional model for studying the transport phenomena during the PSA process in consideration of the heat effect, gas compressibility, radial porosity, and dead volumes. Wang et al. [16] established a 3D hydrodynamic and mathematical model for the RFA in consideration of cross section ratio of central channel to outer channel, and the opening ratios of the central channel and the outer channel. They found that radial inward flow was proved to have a better performance than radial outward flow, and the π-type of inward flow is a little better than that of the z-type of inward flow. Hamedi et al. [17] found a novel radical flow reactor with lower pressure drop, and compared the optimized reactor with an optimized conventional axial flow reactor to ascertain the superiority of the proposed reactor configuration. Yang et al. [18] performed a numerical simulation of a PSA oxygen production process based on radial-flow adsorber and found that the centripetal π-flow radial adsorber has the best flow characteristics for the PSA process.

Most of the above studies on RFA mainly focused on gas flow distribution and internal structure efficiency. There are limited experimental verification on PSA oxygen production. Chiang et al. [19] verified a new PSA system with radial flow bed on oxygen production experimentally under a cycle time less than 30 s. It has been showed that enriched oxygen could be produced when air was fed inward. Huang et al. [20] established a radial flow rapid pressure swing adsorption (RPSA), and found

that small adsorbent particles and long effective lengths would be better for RPSA. Wang et al. [21] designed a radial flow PSA unit with two-tower configuration for oxygen production in consideration of the gas flow pattern, outer flow channel width and channel structure on oxygen production.

It is very effective to study gas separation by numerical simulation [22–26]. However, most of the studies are based on the 1-D or 2-D RFA [27], and few studies have been reported in research of PSA oxygen production based on 3-D RFA. In actual PSA processes, the gas separation in the adsorption bed is a coupling process of multiple complicated processes such as gas flow, heat transfer and mass transfer. In order to investigate the characteristic of gas flow during PSA for oxygen production and improve oxygen production efficiency, a 3-D gas–solid two-phase PSA model was established for the π-shaped centripetal RFA (CP-π RFA). The flow and heat transfer characteristics, oxygen distributions, effect of product flow rate and cycle times were comparatively studied using this model. This model can better reflect the actual PSA oxygen production process. The laws of the pressure, velocity, temperature and oxygen distribution in the adsorber could provide an important technical reference for CP-π RFA in the PSA for oxygen production.

## 2. Experiments

### 2.1. Materials

The PSA experiment was carried out with one laboratory-scaled apparatus as shown in Figure 2. The LiX zeolite (CECA-G5000) were purchased from Luoyang Jian long Micro-Nano Novel Materials Co., Ltd. (Beijing, China), whose properties are shown in Table 1. The adsorption isotherms of pure $N_2$ and $O_2$ in the LiX zeolite were measured at two different temperatures (278 K and 293 K) up to 350 kPa by volumetric apparatus, after the zeolite was vacuum heated at 623.15 K in 12 h.

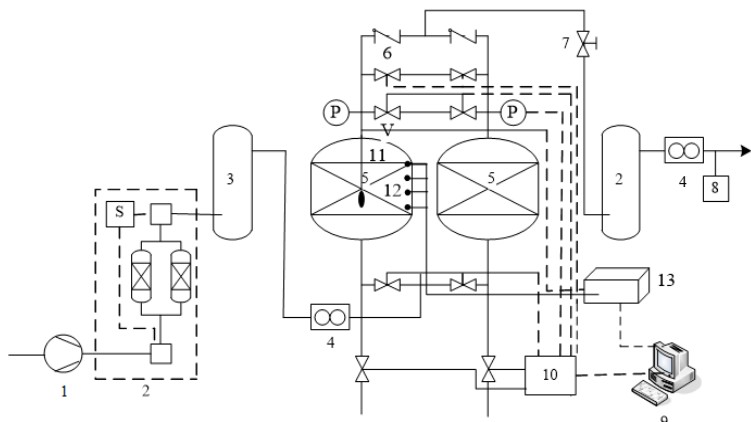

**Figure 2.** Schematic of the pressure swing adsorption (PSA) experimental setup. 1—air compressor, 2—pretreatment system (alumina dehydration), 3—feed gas tank, V-solenoid valve, P—pressure sensor, 4—mass flow controller, 5—radial flow adsorber, 6—check valve, 7—throttle valve, 8—Zirconia oxygen analyzer, 9—computer, 10—PLC, 11—differential pressure sensor, 12—pressure sensor, 13—data acquisition system.

**Table 1.** Parameters of adsorbent particle and adsorbent column.

| Adsorbent | Values | Adsorbent Column | Values |
|---|---|---|---|
| Adsorbent | LiX$_{(CECA-G500)}$ | Bed external porosity | 0.4 |
| Type | Sphere | Dispersion coefficient ($m^2 \cdot s^{-1}$) | $5 \times 10^{-5}$ [28] |
| Particle radius (mm) | 0.8 | LDF constant for oxygen, $k_{O2}$ ($s^{-1}$) | 62.0 [29] |
| Particle density (kg·m$^3$) | 1035 | LDF constant for nitrogen, $k_{N2}$ ($s^{-1}$) | 19.7 [29] |
| Heat capacity (J·kg$^{-1}$·K$^{-1}$) | 1100 | Ambient temperature (K) | 298 |
| Thermal conductivity (W·m$^2$·K) | 0.2 | | |

## 2.2. Experiment Process

The experimental gas was supplied into the RFA with the mass flow of 2.2 m$^3$·h$^{-1}$ (20 °C, 101.3 kPa) through the pretreatment system by the ZW700A-20/B compressor, where each 2.133 kg of CECA-G5000 was loaded. The continuous circulation of oxygen and nitrogen separation process was taken control of the solenoid valve switch through PLC (Programmable Logic Controller). In the experiment, the gas flow rate and the pressure of adsorbent columns were respectively measured by the mass flow meter and the pressure sensor (MPX5700D), whose results were stored in the data acquisition card and converted into the computer. Oxygen content was detected by zirconia oxygen analyzer (ZO·101T type).

## 3. Numerical Model

### 3.1. Physical Model

The main structure of the CP-π RFA consists of a central channel, a central distribution hole, an adsorbent layer, an outer distribution hole and an outer channel, as shown in Figure 3a, with the parameters listed in Table 2. The physical model was established with 1/4 part of RFA as the calculation area, since the cylindrical axisymmetric structure with two symmetry planes, as shown in Figure 3b. The schematic of the CP-π RFA including the important size are shown in Figure 3c. The origin of coordinates was set on the center of the end of the air inlet of the adsorber, with the X-axis and Z-axis along the radial direction and axial direction of the cylindrical structure. The adsorbent layer was full filled with the adsorbent in the spherical shape of 1.6 mm diameter, and the adsorption bed porosity is 0.4.

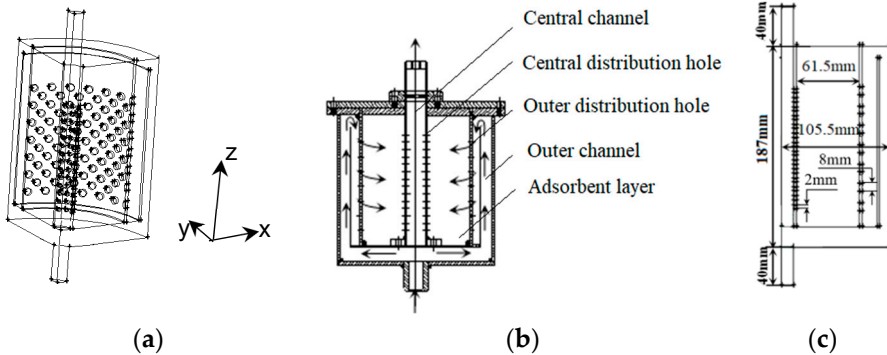

|          (a)          |          (b)          |          (c)          |

**Figure 3.** Physical model of the radial flow adsorber of centripetal RFA (CP-π RFA), (**a**) structure of the CP-π RFA, (**b**) physical model of the CP-π RFA, (**c**) the schematic of the CP-π RFA.

**Table 2.** Parameters of the CP-π RFA.

| Parameter | Values | Parameter | Values |
|---|---|---|---|
| Molar feed composition | O$_2$-N$_{2(21–79\%\ by\ volume)}$ | Inner diameter of adsorber (mm) | 211 |
| Diameter of adsorber (mm) | 219 | Adsorption bed width (mm) | 61.5 |
| Sorbent filling height (mm) | 187 | Outer channel width (mm) | 25 |
| Entrance radius (mm) | 12 | Central channel width (mm) | 13 |
| Entrance length (mm) | 40 | Flow distributor width (mm) | 3 |
| Opening ratio of outer channel (%) | 27% | Diameter of central channel hole (mm) | 2 |
| Opening ratio of central channel (%) | 17% | Diameter of outer channel hole (mm) | 8 |
| Vent length(mm) | 40 | Vent radius (mm) | 12 |

In addition, it is convenient to combine O$_2$ and Ar species into one pseudo species O$_2$-Ar and treat air as a binary mixture of N$_2$ and O$_2$ with a mole fraction of 79: 21 for calculations in the model, because of almost the same capacity of O$_2$ and Ar in the experiment adsorbent.

### 3.2. Mathematical Model

The PSA separation process includes four stages: co-current pressurization (Pr) with feed gas, selective adsorption (Ad) of $N_2$ from a compressed air stream to produce an oxygen enriched gas, countercurrent blowdown (Cb) of column to desorption pressure and purge (Pu). In order to establish a mathematical model of the molecular sieve adsorption process for nitrogen-oxygen mixed gas, the actual PSA process [30] must be simplified as the process followed.

Mass conservation equation:

$$\frac{\partial \varepsilon \rho_f}{\partial \tau} + \nabla\left(\varepsilon \rho_f \cdot u\right) = -S_m \tag{1}$$

$$S_m = \sum_{i=1}^{n} S_i \tag{2}$$

The mass balance equation for component $i$ [25]:

$$\frac{\partial}{\partial \tau}\left(\varepsilon \rho_f Y_i\right) + \nabla \cdot \left(\varepsilon \rho_f u Y_i\right) - \nabla \cdot \left(\varepsilon \rho_f D_i \nabla Y_i\right) = -S_i \tag{3}$$

$$S_i = (1-\varepsilon)\rho_p M_i \frac{\partial q_i}{\partial \tau} \tag{4}$$

According to the momentum balance, the porous media are modeled by the addition of a momentum source term to the standard fluid Navier–Stokes equations (laminar flow). The modified N-S equation is written as:

$$\frac{\partial}{\partial \tau}\left(\rho_f u_i\right) + \nabla \cdot \left(\rho_f u u_i\right) = \rho_f g - \nabla p + \mu \nabla^2 u + \frac{\mu}{3}\nabla(\nabla \cdot u) + S_v \tag{5}$$

In the porous-media zone, apart from the usual momentum sources, there would be momentum changes in the bed due to gas sorption, which was not considered in the previous relevant studies. Hence, an additional source term is added in the momentum equation and the momentum source term is as follows:

$$S_v = -\left(\mu \frac{u}{\alpha} + C_2 \rho_f |u| u + S_m u\right) \tag{6}$$

In this study, coefficients $\alpha$ and $C_2$, which are given by the relations developed by Ergun for flow in a packed bed, are as follows:

$$\alpha = \frac{\varepsilon^3 d_p^2}{150(1-\varepsilon)^2} C_2 = 1.75 \frac{1}{d_p} \frac{1-\varepsilon}{\varepsilon^3} \tag{7}$$

According to the energy balance, for an adiabatic bed with no heat transfer with the surroundings, the overall heat balance may be written as follows.

Gas energy Equation (8):

$$\frac{\partial}{\partial t}\left(\varepsilon \rho_f e_f + (1-\varepsilon)\rho_p e_p\right) + \nabla \cdot \left(\vec{v}\left(\rho_f e_f + p\right)\right) = \nabla \cdot \left[k_{eff}\nabla T + \left(\overline{\overline{\tau}} \cdot \vec{u}\right)\right] + S_f^h \tag{8}$$

$$S_f^h = (1-\varepsilon)\rho_p \sum_i \left(-\Delta H_i \frac{\partial q_i}{\partial t}\right) \tag{9}$$

The effective thermal conductivity is taken as:

$$k_{eff} = \varepsilon k_f + (1-\varepsilon)k_p \tag{10}$$

Solid energy equation:

$$\left(\rho_p C\right)s\frac{\partial T_p}{\partial \tau} = \nabla \cdot \left(k_s \nabla T_p\right) - \frac{6h_f}{d}\left(T_p - T_f\right) + \rho_p \Sigma_i \left(-\Delta H_i \frac{\partial \overline{q}_i}{\partial \tau}\right) \tag{11}$$

The equation of state:

$$\rho_f = \frac{p}{\frac{R}{M_w}T} \tag{12}$$

Mass transfer rate [31,32]:

$$\frac{\partial q_i}{\partial \tau} = k_i(q_i^* - q_i) \tag{13}$$

$$k_i = \frac{\Omega D_e}{R_p^2} \tag{14}$$

Adsorption equilibrium:

$$q_i^* = \frac{K_i p_i}{1 + \sum\limits_{k=1}^{N} b_k p_k} K_i = k_1 \exp\left(\frac{k_2}{T}\right) b_k = k_3 \exp\left(\frac{k_4}{T}\right) \tag{15}$$

The isotherm parameters for LiX zeolite are listed in Table 3.

**Table 3.** Adsorption isotherm parameters.

| Adsorbate | $k_1$ (mol·kg$^{-1}$·Pa$^{-1}$) | $k_2$ (K) | $k_3$ (Pa$^{-1}$) | $k_4$ (K) | $\Delta H$ (kJ·mol$^{-1}$) |
|-----------|---------------------------------|-----------|--------------------|-----------|----------------------------|
| $O_2$ | $7.87 \times 10^{-9}$ | 1541.211 | $6.79 \times 10^{-10}$ | 1968.24 | 12 |
| $N_2$ | $9.86 \times 10^{-9}$ | 2010.908 | $1.67 \times 10^{-9}$ | 2250 | 18 |

The adsorption isotherm of CP-π RFA was confirmed to follow the Langmuir relationship in Figure 4. These isotherms were measured in our laboratory apparatus by using a high-pressure volumetric apparatus up to 350 kPa, where the adsorption isotherms of pure $O_2$ and $N_2$ on LiX zeolite were measured at two different temperatures (278 K and 293 K).

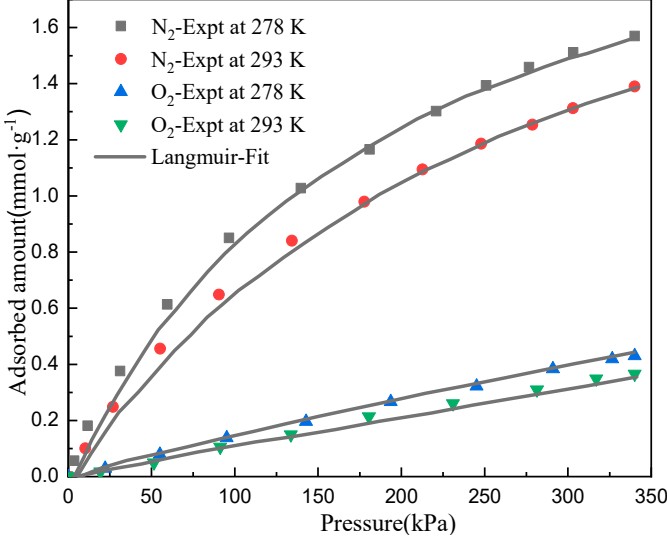

**Figure 4.** Adsorption isotherms for pure $O_2$ and $N_2$ on LiX zeolite.

### 3.3. Initial and Boundary Conditions and Model Parameters

The Skarstrom PSA cycle was employed comprising four basic steps named as pressurization with air (Pr), adsorption (Ad), countercurrent blowdown (Cb), and low pressure purge (Pu). The Skarstrom cycle in its basic form utilizes two packed adsorbent beds, which both undergo these four operations alternately. The initial conditions of the PSA oxygen production process of the CP-π RFA are shown in Table 4. The Skarstrom cyclic sequences and the boundary conditions are summarized in Table 5. The inlet of the bed is set to the mass flow rate boundary condition, and the pressure-flow curve formula of the compressor is:

$$Q = 7.23 - 5.633 \times 10^{-6}\, p \tag{16}$$

In the following, we take Adsorber 1 as an example to explain the setting of the entrance and vent boundary conditions. The PSA process starts from the Pr step of Adsorber 1. During the Pr process, the vent of Adsorber 1 is closed, and air flows into the CP-π RFA at a certain mass flow rate, and the pressure in Adsorber 1 is constantly rising. After the Pr process has completed, the valve at the vent is opened and product gas is obtained. The high operating pressure in Pr process is not an input operating pressure, but is determined by other process variables. During the Ad process, the pressure at the vent is set to the pressure value at the end of the Pr step. At the end of Ad process, the product vent is closed and the feed is turned to Adsorber 2. During the Cb process, the pressure at the feed end is set to 101,325 Pa. When the pressure in Adsorber 1 drops to a lower pressure value, the Cb process is considered to complete. During the Pu process, the product gas of Adsorber 2 is used to purge Adsorber 1, and the pressure at the feed end of Adsorber 1 is kept at 101,325 Pa. The purge mass flow rate is determined by the product recovery.

**Table 4.** Initial conditions.

| Parameter | Values |
|---|---|
| Pressure (Pa) | 101,325 |
| Air-temperature (K) | 298 |
| Solid-temperature (K) | 298 |
| Mass fraction of $O_2$ in the gas-phase | 0.233 |
| Amount of $O_2$ adsorbed per unit mass of sorbent (mol·kg$^{-1}$) | 0.0262832 |
| Amount of $N_2$ adsorbed per unit mass of sorbent (mol·kg$^{-1}$) | 0.6328067 |

**Table 5.** Skarstrom cyclic sequence and boundary conditions for Adsorber 1.

| Step | Schematic Diagram | Duration(s) | Boundary Conditions for Adsorber 1 | | | |
|---|---|---|---|---|---|---|
| | | | Entrance | Vent | Adsorber Wall | Adsorber Axis |
| Pr Step | | 7 | mass flow inlet | wall | wall | axis |
| Ad Step | | 5 | mass flow inlet | pressure outlet | wall | axis |
| Cb Step | | 3 | pressure outlet | wall | wall | axis |
| Pu Step | | 5 | pressure outlet | mass flow inlet | wall | axis |

### 3.4. Meshing and Method of Solution

The mathematical model described above was realized in computational fluid dynamics (CFD) software Fluent using the control volume method. Prior to the CFD calculation, the geometry with its structured grid was generated as shown in Figure 5. There are 230 nodes non-uniformly distributed along the column height and 51 nodes non-uniformly distributed along the radial direction with high grid resolution at the near wall region, resulting in about 11,730 quadrilateral cells. CFD simulations were performed using a double-precision unsteady-state implicit solver, with the coupled algorithm to solve the pressure-velocity coupling problem in the momentum equations. As for the implementation of adsorption in this study, the UDS (user-defined scalars) and UDF (user-defined functions) were applied to program the mass transfer rate Equation (1) and the source terms, etc. (Equations (2), (4), (6), (8), (11), and (14)). A PRESTO (pressure staggering option) discretization scheme was used for the pressure [33–35], whereas a quick scheme for the discretization of density. To reduce numerical diffusion, a second-order upwind scheme was selected for the discretization of the momentum and energy equations.

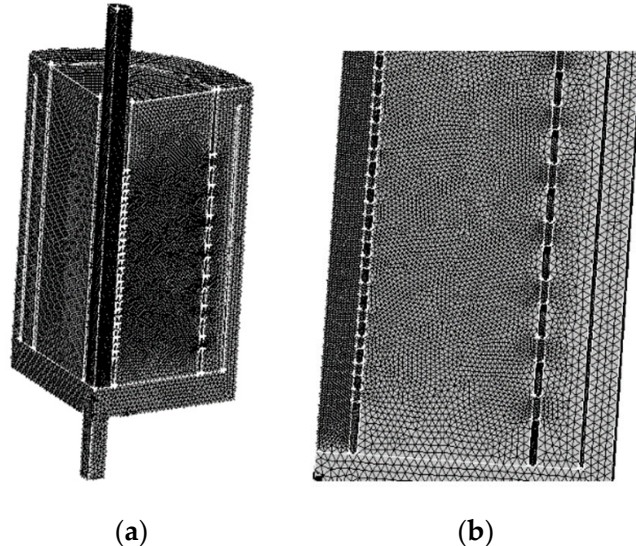

(**a**)　　　　　　　　　　　　　　　　　(**b**)

**Figure 5.** Model grid: (**a**) overall meshing, (**b**) partial encryption.

## 4. Results and Discussion

### 4.1. Model Validation

The flow and heat transfer process in adsorption and separation processes are difficult to verify by experiments. Usually, the product concentration or pressure during the experimental cycle is used to verify the accuracy of the model. This article only uses the pressure verification method to verify the model. Figure 6 shows the results of pressure change in the outer channel of CP-π RFA at the cycle steady state (after the eighth cycle) from the experimental and numerical methods. The cycle steady state means that the oxygen mole fraction of product gas at the outlet of the CP-π RFA gradually reaches a steady state. Satisfactory agreements are obtained at different times during the whole cycle with the same trends and small overall error range. Comparing this with the experimental results, the PSA oxygen two-phase model of CP-π RFA established in this paper is accurate enough to simulate the adsorption process and results. Therefore, this mathematical model is used in this study to describe the flow and heat transfer characteristics in the CP-π RFA [36,37].

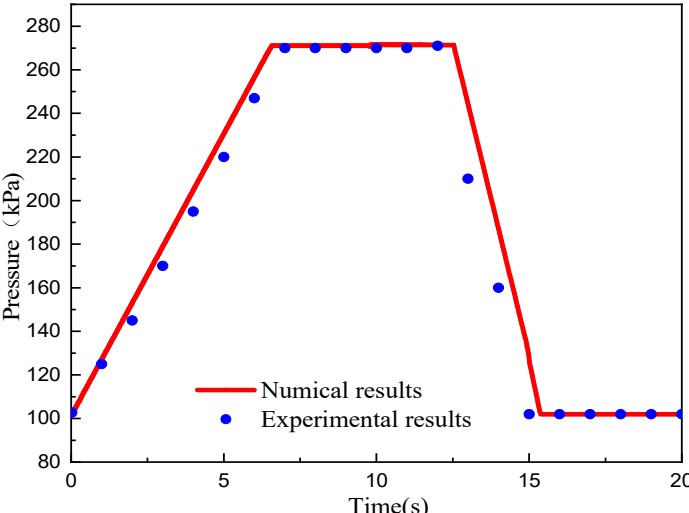

**Figure 6.** Pressure changes in the outer channels of CP-π RFA at the cycle steady state.

### 4.2. Flow and Heat Transfer Characteristics of CP-π RFA

The distribution of pressure in the adsorbent bed at the cycle steady state (after the eighth cycle) for the CP-π RFA are shown in Figures 7 and 8. During the Pr process, the pressure keeps rising, reaching to the highest point 272.265 kPa. In the initial stage of Ad step, there is an obvious peak of pressure near the outer hole. The number of gas pressure changes a lot with the process of source gas through the adsorber and a result of $N_2$ caught in the adsorbent bed, while the pressure in the adsorber increases slightly at the end of Ad step. Finally, the highest pressure in the bed is 271.353 kPa. At the end of Cb step, the pressure in the adsorber drops to 101.364 kPa, due to the resistance of the adsorbent layer. After the Cb step, the pressure in the Pu step is relatively stable, which decreases to 101.325 kPa in the major region.

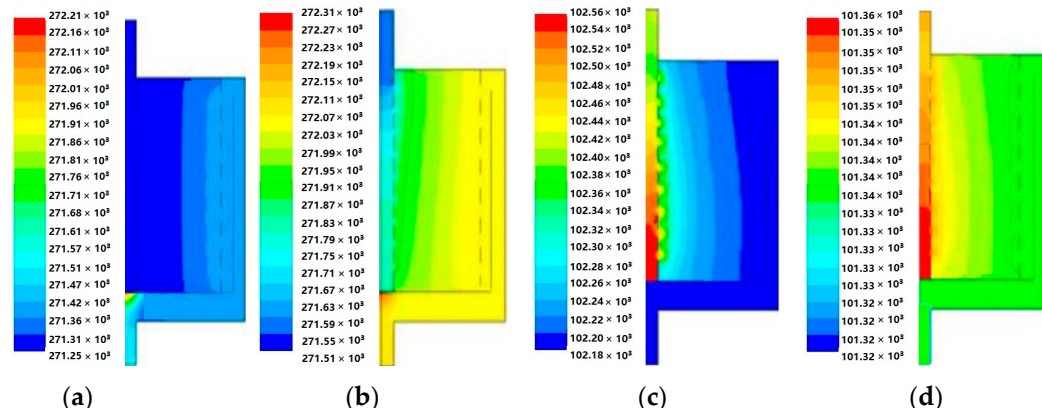

**Figure 7.** Pressure changes in the CP-π RFA at the end of steps of the cycle steady state: (**a**) Pr, (**b**) Ad, (**c**) Cb, (**d**) Pu.

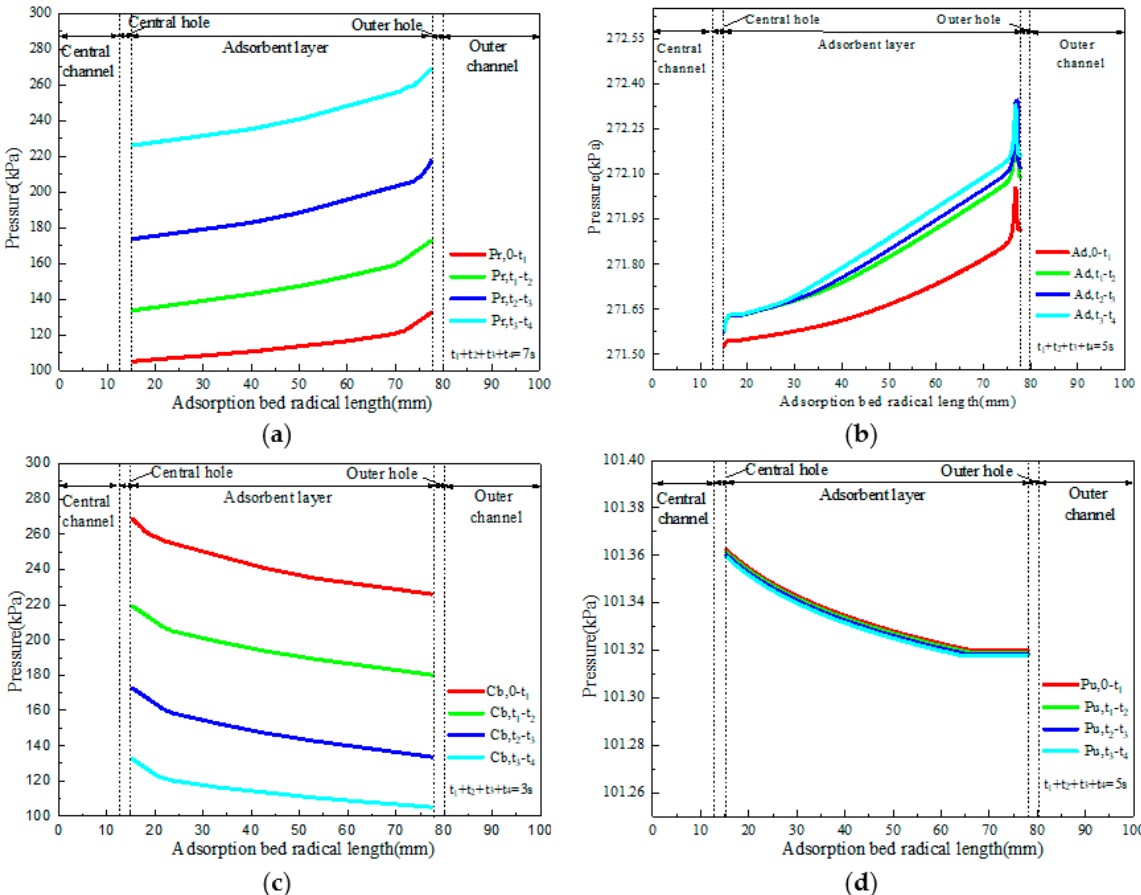

**Figure 8.** Pressure profiles in the CP-π RFA in the process of each step of the cycle stead state at the axial central position: (**a**) Pr, (**b**) Ad, (**c**) Cb, (**d**) Pu.

Figure 9 shows velocity profiles in the adsorbent bed during the process of each step of the cycle steady state for the CP-π RFA. During the Pr step, the gas flow velocity near the outer hole is very high because of the gas enters the adsorber from the outer channel, then due to the resistance of the adsorbent layer, the gas flow velocity decreases from 10.5 m·s$^{-1}$ to 2.2 m·s$^{-1}$. At the Ad step, the gas flow velocity in the adsorbent layer are smaller than those near the outer hole and central hole, which are 5 m·s$^{-1}$ and 2.5 m·s$^{-1}$, respectively. At the Cb step, there is a large pressure reduction on both sides during the initial Cb step. The gas flow velocity near the central hole reaches the peak, about 62 m·s$^{-1}$. This is because the strongly adsorbed components are desorbed from the adsorbent, so the gas flow velocity near the central channel becomes large. With the decrease of the pressure drop between the two sides, the flow velocity of the desorbed gas decreases gradually, same as the radial velocity. At the Pu step, the gas flow velocity in the adsorber gradually decreases along the radial length.

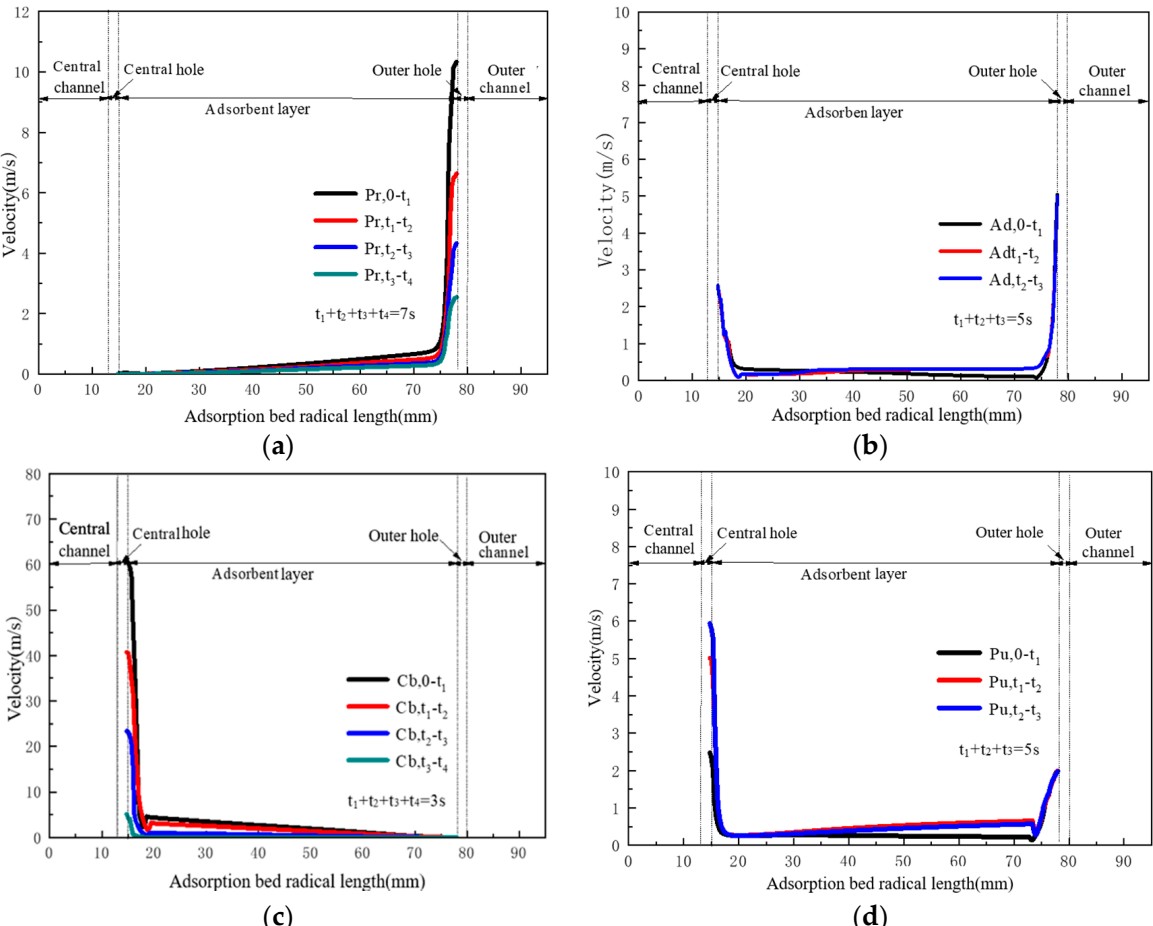

**Figure 9.** Velocity profiles in the CP-π RFA in the process of each step of the cycle steady state at the axial central position:(**a**) Pr, (**b**) Ad, (**c**) Cb, (**d**) Pu.

Figure 10 shows the temperature profiles in the adsorbent layer at the end of the four steps of the cycle steady state for the CP-π adsorber [38]. It is clearly observed that the temperature of the gas–solid two phases at the end of the four steps are all increased and the temperature in the outer hole are all higher than those in the central hole. During the Pr step, the temperature difference between the gas–solid two phases gradually decreases from about 4 K near the outer hole to 2 K near the central hole, while the radial temperature difference in the bed reaches 15 K after the Pr step. During the Ad step, the temperature of the gas–solid two phases in the bed is increased, and the temperature reaches 287 K. During the Cb step, the temperature of the adsorbent decreases from 294 K to 277 K, and the gas phase temperature in the bed is reduced from 304 K in the adsorption state to about 293 K, due to heat transfer during gas desorption. During the Pu step, the temperature of the adsorbent layer decreases from 307 K to 287 K, and the gas phase temperature in the adsorber is reduced from 296 K to 280 K.

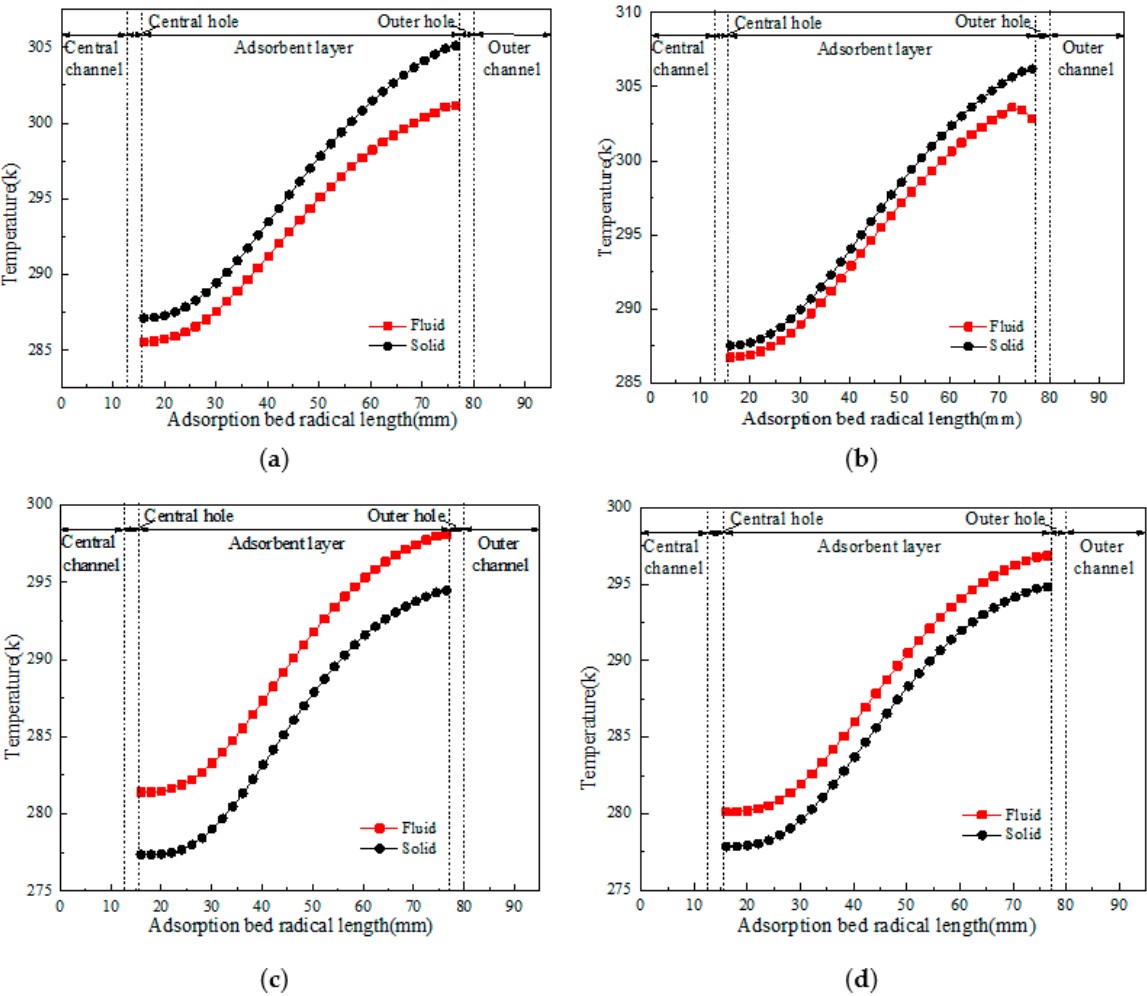

**Figure 10.** Comparison of temperature profiles in the CP-π RFA at the end of the four steps of the cycle steady state at the axial central position:(**a**) Pr, (**b**) Ad, (**c**) Cb, (**d**) Pu.

### 4.3. Oxygen Distributions of CP-π RFA

Figure 11 shows the oxygen distribution at the end of each step in the cycle steady state for the CP-π RFA. The mass transfer area is further shifted to the outlet of the adsorber, the oxygen purity of 99% and 98.6% can be obtained in the outlet after the Pr and Ad steps, while those of 85.4% and 98.4% can be obtained in the outlet after the Cb and Pu steps.

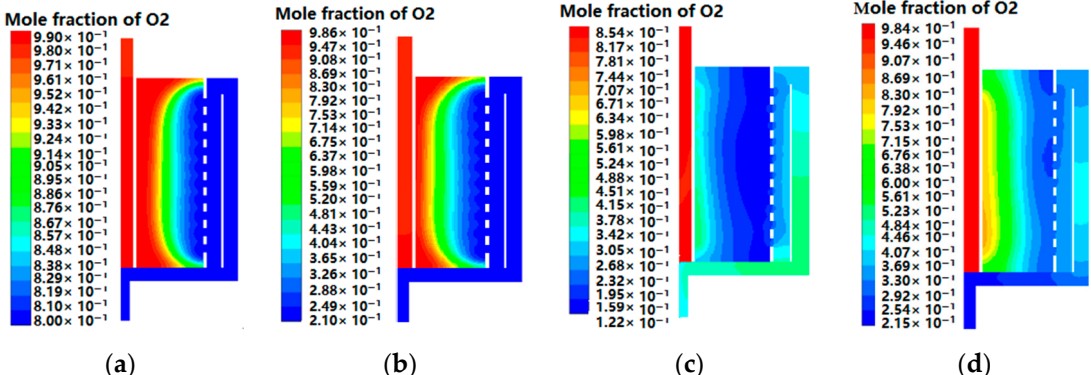

**Figure 11.** Oxygen distribution in the CP-π RFA at the end of four steps of the cycle steady state: (**a**) Pr, (**b**) Ad, (**c**) Cb, and (**d**) Pu.

Figure 12 shows the oxygen purity profiles at the axial central position at the end of each step in the cycle steady state for the CP-π RFA. After the Pr and Ad steps, the mass transfer area of oxygen is compressed with a result that the oxygen purity is 98.6%, and highly concentrated oxygen flows out from the outlet of CP-π RFA. At the Cb step, amounts of nitrogen are desorbed from the adsorbent layer and flow out from the inlet of CP-π RFA, leading to the lower oxygen purity in the outer hole of 12.2%. While the oxygen purity in the central channel is still as high as 84.8%. At the end of Pu step, the oxygen purity near the central hole of CP-π RFA reaches up to 83%, and the higher oxygen purity kept in the central runner is beneficial to the oxygen production for the next cycle.

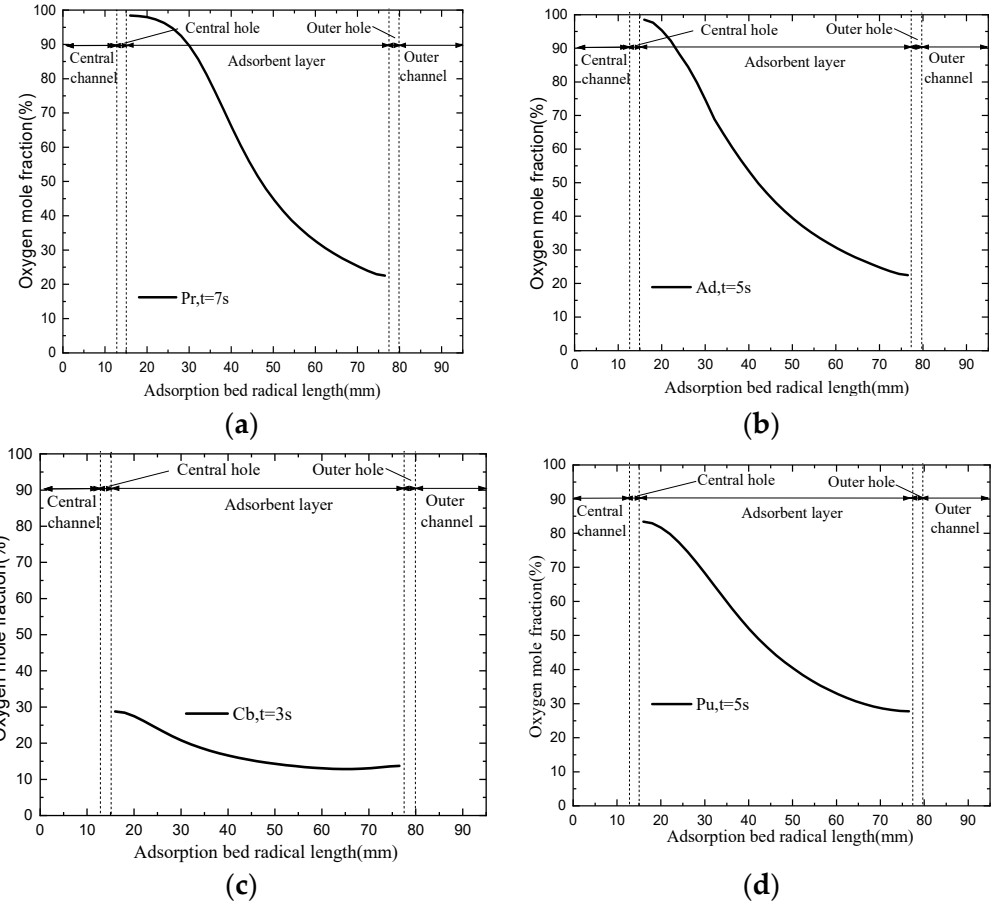

**Figure 12.** Comparison of axial $O_2$ purity profiles at the end of the 4 steps in the cycle steady state for the CP-π RFA:(**a**) Pr, (**b**) Ad, (**c**) Cb, (**d**) Pu.

*4.4. Effect of Product Flow Rate and Cycle Times on Oxygen Production Performance of CP-π RFA*

The oxygen product flow rate will affect the oxygen purity and recovery. Figure 13 shows the changes in oxygen product purity with the cycle times and the oxygen product flow rate for the CP-π RFA. The oxygen product purity increases from 66% to 98.6% significantly after the eighth cycle, and since then enters the stable cycle process. The oxygen product purity decreases from 98.6% to 78.5%. Under the same conditions, increasing the product flow rate requires increasing the feed airflow, which will further increase the airflow speed and the mass transfer movement in the adsorber. In addition, due to the constant mass transfer coefficient of the adsorbent, the higher product flow rate prolongs the mass transfer area and reduces the oxygen purity.

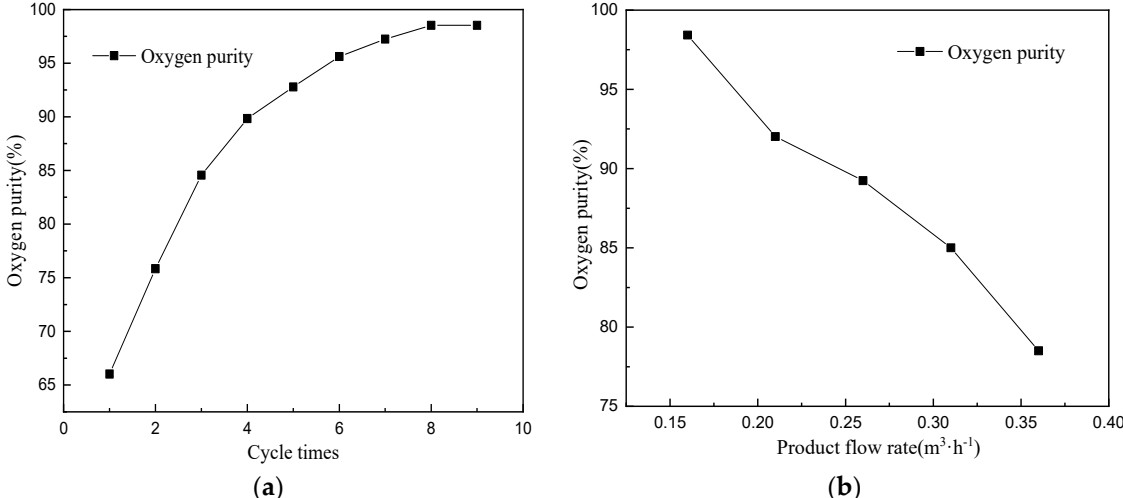

**Figure 13.** Changes of cycle times and product flow rate for the CP-π RFA: (**a**) changes in oxygen purity with cycles times, (**b**) changes in oxygen purity with product flow rate.

The oxygen product flow rate has an important influence on the oxygen production performance. By lowering the oxygen product purity, it will generate more oxygen which can decrease operational costs, reduce the equipment size, and save the initial investment. For instance, oxygen product purity of 80% can meet a lot of industrial requirements.

Compared with the traditional ones, the RFA is convenient for large-scale, has low pressure resistance, and saves energy. The concentration of PSA oxygen production can reach 95% [39]. Generally, the product concentration is adjusted according to user needs. In many cases, the oxygen concentration is set to 90%. Different types of molecular sieves have different productivity. Industrial oxygen molecular sieves are designed to be 80–100 $m^3$/h through the conversion of vacuum swing adsorption (VSA), and oxygen productivity of PSA is lower than VSA [40].

## 5. Conclusions

A 3-D model for CP-π RFA has been established and tested with experimental results. The simulated results show the flow, heat transfer characteristics, and oxygen distributions in the CP-π RFA. Simulated results allow the following conclusions:

In the initial stage Ad step, there is a distinct pressure peak near the outer hole. In addition, the raw material gas enters the adsorber and $N_2$ is adsorbed by adsorbent bed, which causes the pressure change obviously, and the pressure in the adsorber increases slightly at the end of the Ad step.

In the Pr step, the gas flow velocity decreases to 2.2 m·s$^{-1}$ near the outer hole. In the Ad step, the gas flow velocity near the outer hole and the central hole are 5 m·s$^{-1}$ and 2.5 m·s$^{-1}$, respectively. The maximum speed near the central hole is about 62 m·s$^{-1}$ and 6 m·s$^{-1}$ in the Cb and Pu step.

In a stable cycle, the temperature of adsorbent bed in the CP-π adsorber changes significantly. The radial temperature difference in the adsorbent bed reaches 15 K after the Pr step, and the temperature increases to 287 K after the Ad step. The temperature of the adsorbent decreases to about 277 K in the Pu step.

The oxygen purity is achieved to be 98.6% after the Ad step, and the oxygen purity in outlet of bed can reach up to 84.8% with purging by highly concentrated product in the end of Pu step.

This model can be used in multicomponent PSA process. The pressure changes during the Pr step and Cb step are calculated by the other process variables, not semi-empirical formulas. This work could provide a reference for the application of CP-π RFA and improve the design of CP-π RFA. In the future, detailed research will be mainly focused on the optimization of structural parameters and industrial applications.

**Author Contributions:** Conceptualization, H.W. and X.Y.; methodology, H.W., C.Z., X.Y. and Z.L.; writing, H.W., and X.Y.; project administration, Y.L., C.L., Y.H., X.M. and H.W. All authors have read and agreed to the published version of the manuscript.

**Funding:** This research was funded by Beijing Natural Science Foundation (No. 8182019), National Natural Science Foundation of China (No.51578065), Beijing Education Commission General Project (No.KM202011417007), Premium Funding Project for Academic Human Resources Development in Beijing Union University.

**Conflicts of Interest:** The authors declare no conflict of interest.

## Nomenclature

| | |
|---|---|
| $d_p$ | Diameter of adsorbent particle, m |
| $D_i$ | Mass dispersion rate, $m^2 \cdot s^{-1}$ |
| $e_f$ | Total fluid energy, $J \cdot kg^{-1}$ |
| $e_p$ | Total solid medium energy, $J \cdot kg^{-1}$ |
| $\Delta H$ | Heat of adsorption, $kJ \cdot mol^{-1}$ |
| $K_i$ | Langmuir parameter, $mol \cdot kg^{-1} \cdot kPa^{-1}$ |
| $k_1$ | Langmuir parameter, $mol \cdot kg^{-1} \cdot kPa^{-1}$ |
| $k_2, k_4$ | Langmuir parameter, K |
| $k_3$ | Langmuir parameter, $kPa^{-1}$ |
| $k_{eff}$ | Effective bed thermal conductivity, $W \cdot m^{-2} \cdot K$ |
| $k_i$ | Mass transfer constant, $s^{-1}$ |
| $M_i$ | Molar weight of component $i$, $kg \cdot mol^{-1}$ |
| $p$ | Gas pressure, Pa |
| $p_i$ | Partial pressure of component $i$, Pa |
| $q_i$ | Solid-phase adsorbate concentration, $mol \cdot kg^{-1}$ |
| $q_i{}^*$ | Adsorbate concentration in equilibrium with gas phase, $mol \cdot kg^{-1}$ |
| $Q$ | Volume flow rate, $m^3 \cdot s^{-1}$ |
| $S_i$ | Mass source term of the component $i$, $kg \cdot m^{-3} \cdot s^{-1}$ |
| $S_m$ | Total mass source term, $kg \cdot m^{-3} \cdot s^{-1}$ |
| $S_v$ | Momentum source term, $N \cdot m^{-3}$ |
| $t$ | Adsorption time, s |
| $T$ | Temperature, K |
| $u$ | Velocity vector, $m\ s^{-1}$ |
| $Y_i$ | Mass fraction of component $i$ |
| $\varepsilon$ | Porosity of the fixed bed |
| $\mu$ | Dynamic viscosity of the fluid, $Pa \cdot s$ |
| $\rho_p$ | Density of adsorbent particle, $kg \cdot m^{-3}$ |
| $\rho_f$ | Fluid density, $kg \cdot m^{-3}$ |

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
