# Peer review of "3-D Modeling of Gas–Solid Two-Phase Flow in a π-Shaped Centripetal Radial Flow Adsorber"

_applsci, doi:10.3390/app10020614_

Round 1
Reviewer 1 Report
Adsorption processes for gas separation are considered to be important in the context of the changing raw material and energy supply situation. Highly efficient processes for the provision of oxygen will be increasingly important. Hence, the paper could be an important contribution. However, there are several points that should be addressed prior to publication:
Please check for English language, especially with respect to correct use of articles. Please elaborate on the novelty of this contribution e.g. in comparison to reference [6]. Line 15: adsorber instead of absorber. Line 20: Favourable sounds wrong, as if the model overestimates the capabilities of the adsorber. Lines 23 and 24: Please clarify do you mean that the selected O2 flowrate strongly influences the O2 product concentration? Line 31: What is meant by airspeed? The superficial velocity? Line 56: Novel instead of noval. Line 59: The word generation implies production by means of a chemical reduction. Please replace by production or separation. There are several additional instances of the use of the word generation. Lines 80 to 84: Please shift table 6 forward, so that the relation is clear. What type of isotherms were measured? Single or mixed gas? Please also mention at this point that the measured isotherm could be described by the Langmuir model and that the extended Langmuir equation according to Eq. 15 was employed to describe the adsorption equilibrium. Line 84: What does XXX mean? Line 99: How were dispersion and LDF coefficients in Table 1 determined? Lines 102 and 13: Please add annotations to Figure 2 (a), that enable identification of the elements of the radial bed mentioned in the text. Lines 105 and 106: Add co-ordinate system in to Fig. 2 (b). Line 108: Is this the adsorbent porosity? Line 117: Please identify the geometry, i.e. give a relation between Table 2 and Figure 2 (b). Line 119: It is absolutely necessary to include a nomenclature listing the used symbols and their units of measurement. Line 147, Equation 8: What is E? The sum of internal, kinetic and potential energy? Please include a reference to the source of the equations. Line 169, Equation 16: In (15) q appears to be a loading, in (16) a flowrate. Please try to avoid confusion by using different symbols. Line 175, Table 5: It is absolutely not clear what the intention of this table is. Please identify this clearly, may be by adding an appropriate drawing. Line 197 to 191: What is a PRESTO! discretization scheme? Pressure and density are mentioned, but what schemes were used for composition? This has to be explained more clearly with references where details on the schemes mentioned can be found. Furthermore, what method of integration with respect to time was used and what method was used to describe flow reversal when e.g. moving from Ad to Cb step? How were the interactions of the two beds modelled, i.e. the supply of purge gas from bed 1 to bed 2 in the Ad step and vice versa in the Pu step? Line 193, Figure 3: The figure is not clear. Lines 201 to 203: It is indeed apparent that the pressure is reflected satisfactorily by the proposed model. However, some additional reasoning should be provided as to why it is assumed that this is also true for the flow and the heat transfer. Furthermore, it would be good if the accuracy of the composition prediction could be commented on as well. Line 205, Figure 4: Please correct y-axis legend. Figures 6, 7, 8 and 10: Radial distance instead of radical distance in x-axis caption. Line 208: Is the cyclic steady state meant, when the word stabilised is sued? Please clarify. Page 10, lines 3 to 5: The reason for the high velocities at the in- and outlets should be explained. Lines 272 and 273: This sentence is difficult to understand. Please rephrase. Lines 280 to 293: Some additional information as to why this approach is better that the standard beds should be mentioned.
Author Response
Dear Editors and Reviewers,
I would like to submit the above revised manuscript for your further evaluation for publication in Applied Sciences. I would like to acknowledge the reviewers for providing excellent reviews which are important for us to improve the accuracy and clarify of this manuscript. All comments were considered carefully, and the manuscript was revised accordingly and highlighted in blue color. The point-to-point response to each of the comments is attached below. A supplementary file was added with new figures and tables to incorporate some updated information. The references were renumbered and formatted according to the journal requirement.
I believe that all comments and suggestions have now been fully and satisfactorily addressed in the revised manuscript. Thank you very much for considering our manuscript for potential publication. Please let us know if any further revision is needed.
Sincerely yours,
Haoyu Wang
College of Biochemical Engineering, Beijing Union University, Beijing
No. 18, 3rd District, Fatou Xili, Beijing, 100023, China
Tel: +86-01-52072257

Reviewer 2 Report
The manuscript overall idea is interesting but presents a series of a series of shortcomings that should be addressed. Therefore, I would recommend major reviewers to be done.
A better review about PSA processes should be added, the introduction focus only on RFA.
Dis the authors performed the experiments to identify the isotherms? Where is the isotherms validation? If the authors claim that the experiments were done they should present the experimental data and the isotherms validation.
What the authors mean with the "XXX" in Line 84? - "up to 350 kPa by
84 the XXX, after the zeolite was heated at 573.15 K in 12 hours."
Figure 4, why the authors represented the experimental data with a line?
What is the gains of the approach compared with the traditional ones? What is the purities normally achieved with a PSA? What is the productivity?
Author Response

(The authors gave the same response as above.)

Round 2
Reviewer 2 Report
The authors improved the manuscript. I would now recommend its publication.